# Prosocial and antisocial choices in a monogamous cichlid with biparental care

Shun Satoh [1,2✉], Redouan Bshary [3], Momoko Shibasaki[1], Seishiro Inaba[1], Shumpei Sogawa[1], Takashi Hotta[4], Satoshi Awata [1] & Masanori Kohda[1]

Human society is cooperative and characterized by spontaneous prosociality. Comparative studies on endotherm vertebrates suggest that social interdependence causes the evolution of proactive prosociality. To test the generality of this hypothesis, we modify a prosocial choice task for application to the convict cichlid, *Amatitlania nigrofasciata*, a monogamous fish with biparental care and a strong pair bond. We also affirm that male subjects learn to favor prosocial choices when their mates are the recipients in a neighboring tank. When the neighboring tank is empty, males choose randomly. Furthermore, in the absence of their mates, males behave prosocially toward a stranger female. However, if the mate of the subjects is also visible in the third tank, or if a male is a potential recipient, then subjects make antisocial choices. To conclude, fish may show both spontaneous prosocial and anti-social behaviors according to their social relationships with conspecifics and the overall social context.

[1] Department of Biology and Geosciences, Graduate School of Science, Osaka City University, Osaka, Japan. [2] Department of Evolutionary Studies of Biosystems, The Graduate University for Advanced Studies, Miura, Japan. [3] University of Neuchâtel, Institute of Zoology, Neuchâtel, Switzerland. [4] Department of Psychology, Graduate School of Letters, Kyoto University, Kyoto, Japan. ✉email: symphysodondiscus@icloud.com

P roactive prosociality, the spontaneous unsolicited help of a recipient, is currently rarely documented outside humans. Prosocial behaviors arise from psychological motivation known as "other-regarding preference,"[1] where the immediate reward of the helping action is not material but has been called a "warm glow" in humans[2,3]. Most of the evidence outside humans has been collected using observational and experimental approaches on primates[1,4–8]. In the standard experimental paradigm, the prosocial choice task (PCT), subjects can choose between an antisocial option that rewards only the actor and a prosocial option that rewards both the actor and others[1,4,6,9–11]. If subjects consistently provide food to the recipient[1,4,6], then they are considered to be prosocial, although the help is cost-free in this paradigm.

Early evidence that chimpanzees—a rather cooperative primate— do not show consistent evidence for proactive prosociality in the PCT paradigm[10,12,13] has led to the realization that advanced cognitive abilities may not be the defining prerequisite for the evolution of other-regarding preferences. Instead, comparative studies on a wide range of primates, including lemurs, New World and Old World monkeys, and apes, suggest that allomaternal care underlies prosociality, which is observed in cooperative breeding and biparental care species[7]. In line with this hypothesis, prosocial tendencies have also been reported in non-primate species showing allomaternal care, such as rats[14], dogs[15], and birds (e.g., parrots[16] and magpies[17]). The findings validate that proactive prosociality may be more likely in species with strong social interdependence, which lowers competition between interaction partners whose fitness is linked[4,18,19].

The emerging picture that proactive prosociality is more tightly linked to the social organization of a species and hence its social emotions rather than to advanced cognitive processes suggests that proactive prosociality may also be found in other vertebrate taxa. The so-called social decision-making network in the brain, which comprises both the social behavior network and the mesolimbic reward system, is highly conserved across vertebrate clades, including fishes[20]. In addition, the basic cognitive requirements, such as individual recognition, understanding of social context, and memory[1,5,8], have been documented in various fish species[21–26]. Some fish may also form strong social bonds, such as species forming monogamous pairs that are surrounded by territorial neighbors. Such monogamous fish showing biparental care is hence good candidates for testing the allomaternal care hypothesis[7] and the associated interdependence hypothesis, which states that an individual should show active prosociality toward recipients whose well-being positively affects the direct fitness of self[4,19].

We adjust the PCT for experiments on the monogamous convict cichlid fish *Amatitlania nigrofasciata* (size, 8–10 cm), which engages in the biparental care of young in nests[27–29]. Individuals of this species can visually distinguish between familiar and unfamiliar individuals and change their behavior depending on the social relationship[30]. Few days of mutual courtship involving brushing and quivering movements lead to spawning and a mutual bond, shown by high tolerance within the couple as opposed to high aggressiveness toward other conspecifics[30]. For ~4–6 weeks, the pair guards a small territory around the nest, and both parents dig for food for their young[28] and also attack any approaching conspecifics, egg predators, or predators of young fry[27–31]. Joint reproduction and parental care cause a strong and very direct interdependence between partners with respect to individual fitness. This interdependence and pair bonds can be stable for a long period, i.e., over more than five reproductive cycles in semi-natural tanks (Satoh personal observation). Male subjects can swim into one of two compartments that contain the same amount of food for them (Fig. 1).

Nevertheless, one choice is prosocial in that a visible potential recipient in an adjacent tank would receive the same amount of food. The other choice would yield no food to the potential recipient (Fig. 1). Following Silk et al.[10], we term such a choice "antisocial." We initially focus on testing males with either their established female partner or in the control situation where the neighboring tank is empty. In addition, we expand the test conditions by introducing either another male, a new female, or a new female with the mate of the subject visible for him in the third tank (although the subject cannot provide any food to his mate; Fig. 1).

In this work, we test whether convict cichlids show proactive prosociality toward their reproductive mates. Our key prediction is that if cichlids show proactive prosocial tendencies, then male subjects should choose the compartment that causes food to drop in the adjacent tank more frequently when the female partner is present than in the control condition, wherein the adjacent tank is empty. Moreover, to be considered "proactive," such prosocial choices should be made independently of females showing any indications for soliciting this choice. Thus, we also observe the mate to check for solicitations. For the other three conditions, we expect that the presence of a male would either cause random choices or antisocial choices given that males are competitors. The presence of a solitary new female—with the female partner temporarily removed—could potentially trigger proactive prosocial behavior if male subjects perceive the situation as an opportunity to start a new relationship. By contrast, a solitary new female should either induce random or antisocial choices if the female partner is present, albeit in the second separate tank so that she cannot interfere directly with male choices.

## Results and discussion

**PCT toward female mates.** We first focused exclusively on the comparison between the female mate and the control condition. The frequency of prosocial choices differed significantly between the mate and control experiments but also interacted with an experimental day (generalized linear mixed model (GLMM): $\chi^2 = 70.486$, $p < 0.0001$). Figure 2 illustrates that choices were random during the first 2 days. From day 3 onward, males significantly preferred the prosocial choice when teamed up with their mate, and this preference stabilized from days 6 to 10 (95% confidence intervals overlapping; Fig. 2). Conversely, the choices remained at random levels in the control situation throughout the experiment. The significant difference persisted when we only included males that had been tested in both conditions (Supplemental Fig. S1).

These results are similar to those of a study in which capuchin monkeys were tested with the PCT paradigm[1]. In that study, the monkeys examined the other-regarding preferences. In our subjects, we quantified body orientation at the end of own foraging and turning toward the mate immediately after as the indicators of attention to test whether subjects may monitor the consequences of their choices. The bodies of subjects were not oriented toward the female, and orientation did not differ between the early and late parts of the experiment (cumulative link mixed model (CLMM): $F = 0.230$, $p = 0.632$) (Supplemental Fig. S2). However, males sometimes turned toward their foraging mate, a behavior that was reduced by ~60.8% during days 6–10, i.e., when males had learned to make the prosocial choice. The data yielded a strong tendency that this male monitoring behavior occurred at a higher rate during the early versus later parts of the experiment (binomial GLMM: $\chi^2 = 3.589$, $p = 0.058$) (Supplemental Fig. S2). In addition, we asked whether mates could solicit prosocial choices by male subjects. We also found that the time that the female mate spent in front of the prosocial or antisocial

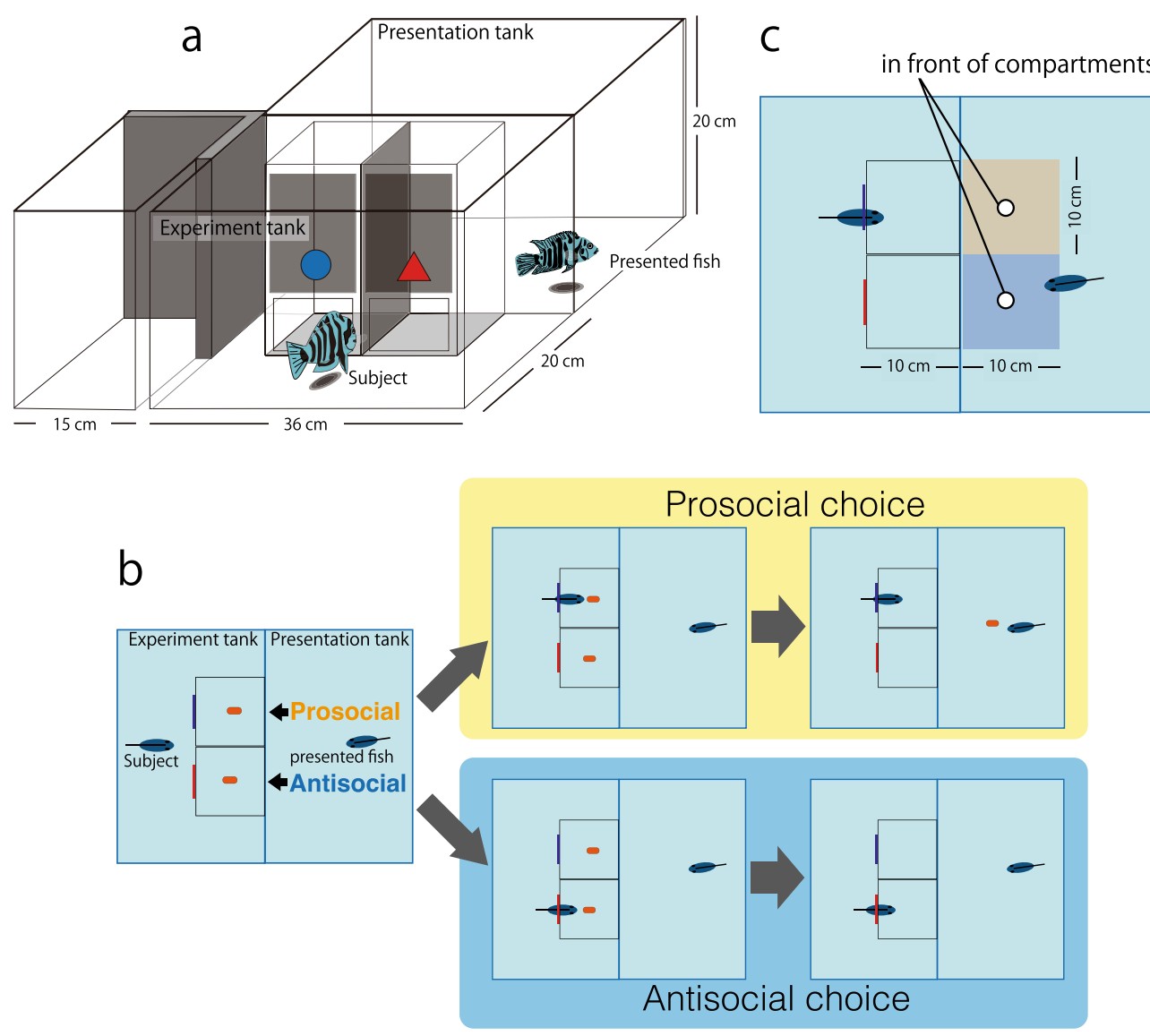

**Fig. 1 Experimental setup of prosocial choice tasks for convict cichlids. a** Schematic of the experimental tank containing two compartments (corresponding to prosocial and antisocial choices), the presentation tank, and the third tank to the left in which the subject's mate could be kept while the potential recipient was a new female. The subject fish were in the experimental tank, and the potential recipients were in the presentation tank; the fish could observe one another directly. **b** Illustration of the choices of either the prosocial or the antisocial compartment and the consequences for the recipient. **c** Illustration of the three possible spatial positions of the subject's mate during trials before subjects made their choice. The mate's head could either be in a zone in front of the prosocial compartment, in a zone in front of the antisocial compartment or distant from either compartment.

choice compartment and her body position were not associated with the choice made by the males (Supplemental Tables S3 and S4). It thus appears to be unlikely that the females solicited prosocial behavior from the males. In sum, these results indicate that males apparently understood the consequences of their prosocial or antisocial choices. These results also support the hypothesis that in this small fish, males favor prosocial choices and show other-regarding preferences toward their mate.

**Effects of social relationship and social situation.** Primatologists considered early on the possibility that prosocial choices may depend on the social relationship of the subjects with the potential recipients (e.g., group members versus unknown individuals)[1,4,6] and social situation (e.g., bystander effects)[32]. Possibly, these social factors strongly affect the intensity of prosociality[1], as with social interdependence[7]. To test this hypothesis, we included additional treatments in our experiment,

inspired by the PCTs performed on primates[1]. If *A. nigrofasciata* only cares about the well-being of interdependent partners, then differences in the prosocial response rate according to social relationships with potential recipients would be expected. Moreover, such differences would allow to reject the alternative interpretation for the mate experiment that subjects generally prefer the option that leads to higher foraging rates.

The full model includes the five treatments and treatment order, as the order was not well balanced (see the Methods section). On the basis of the learning curve shown in the first model (Fig. 2), we only included days 6–10 in the full model. We also found a strong main effect of treatment ($\chi^2 = 138.987$, $p < 0.0001$), but the treatment also interacted with order ($\chi^2 = 21.565$, $p = 0.0058$). Consequently, results in Fig. 3 are shown separately as a function of treatment order. The overall pattern remained rather consistent across treatment order. Combining the independent results for each treatment order to produce the

summary estimates of prosocial tendencies, we asserted that the subjects chose overall randomly in the control treatment ($p = 0.205$). They made significantly above chance prosocial choices when the potential recipient was the mate ($p < 0.0001$) or a new female alone ($p = 0.021$). Contrarily, the males made overall antisocial choices (significantly below chance levels) if the potential recipient was a rival male ($p = 0.0069$) or a new female with the subjects also seeing their mate in the extra nearby tank ($p < 0.0001$) (Fig. 4). These findings confirm that males indeed understood the consequences of their choices and hence provide the first experimental evidence for proactive prosocial motivation in a fish, expressed as other-regarding preferences[1,4]. Kin selection[33,34] can be excluded as the ultimate cause for the males' prosocial choices in this study because the subject and presented fish were unrelated. Instead, in our study species, male fitness is highly interdependent with the fitness of the female partners as

individuals form stable pairs for the biparental care of the brood[27].

Interestingly, the males extended their prosocial behavior toward new females but only in the absence of their mate. Under natural conditions, males treat unfamiliar females as intruders and show aggressive behaviors when the mate is present, as mate switching and/or polygamy is very rare in this species[27]. In our

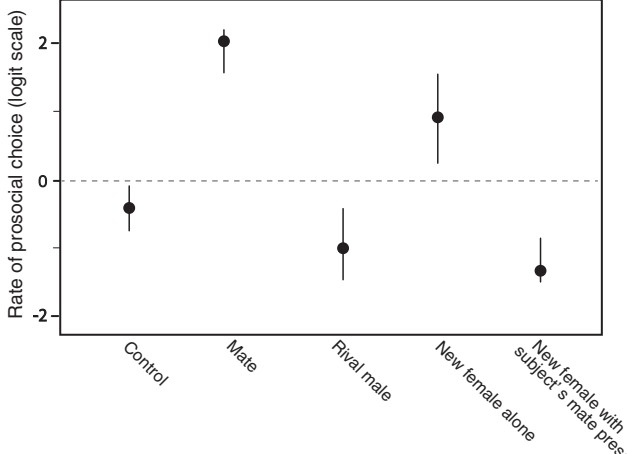

**Fig. 4 Differences in the prosocial choice rate among the control, mate (female), rival male, new female alone, and new female with subject's mate present experiment.** Broken line indicates the expected value of choices. Data are mean ± 95% confidence interval. There are significant differences between expectation of 50% and treatments (male, $p < 0.0001$; rival male, $p = 0.0069$; new female alone, $p = 0.021$; new female with subject's mate present, $p < 0.0001$), whereas there is no significant difference on the control experiment ($p = 0.205$); $p$ values are adjusted using the MVT method.

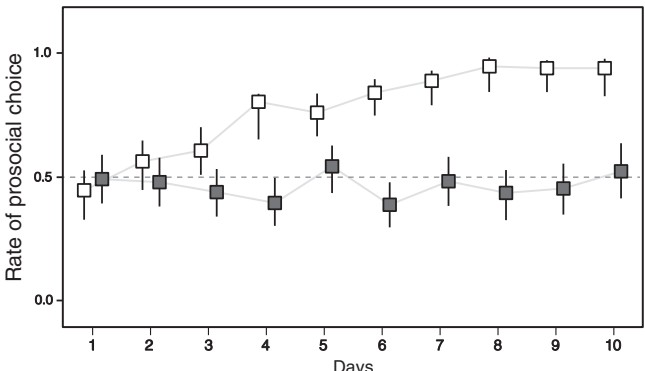

**Fig. 2 Change in the prosocial choice rate during the choice experiment.** White and black squares show the mate experiment ($n = 12$) and the control experiment ($n = 12$) respectively. Broken line indicates an expected value of choices. Data are mean and 95% confidence interval.

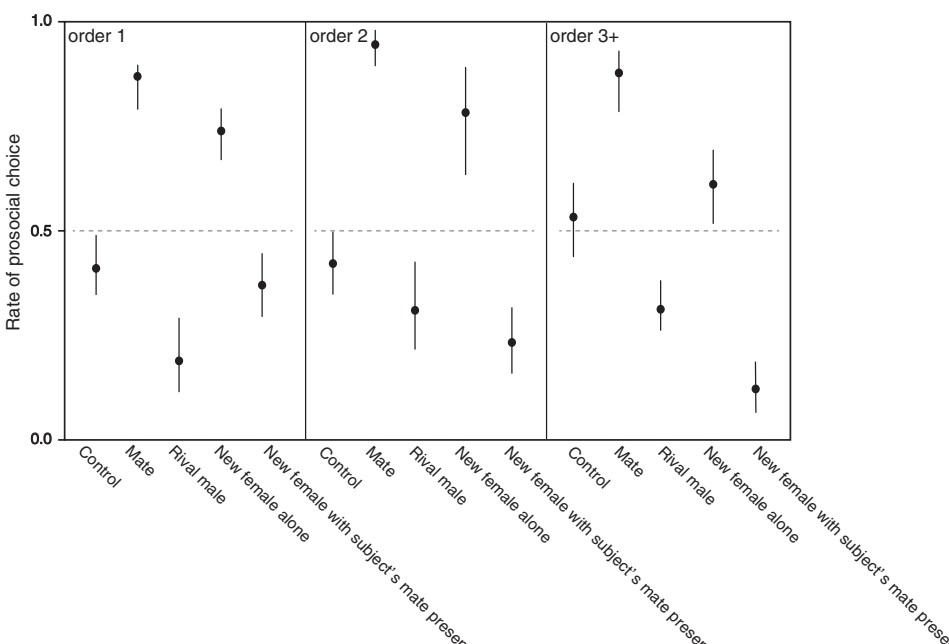

**Fig. 3 Differences in the prosocial choice rate over the last 5 days among treatments in each order.** Data are mean ± 95% confidence interval. Broken line indicates the expected value of choices. Control ($n = 12$), mate ($n = 12$), rival male ($n = 10$), new female alone ($n = 10$), and new female with subject's mate present experiment ($n = 10$) were compared. Given that significant effects on the order of experiments were detected, the rate of prosocial choice is separately shown in every experimental order (see the Results and discussion section).

treatment "new female alone" (as well as in the control and the "rival male" treatments), the subjects were completely isolated (visually and olfactory) from their established mate for the entire duration of the other treatments, i.e., during periods of 10 days each. Under these circumstances, the males possibly viewed the new females as potential future mates and therefore became interested in the well-being of the females. The subjects took several days longer before they preferred the prosocial choice in the "new female alone" treatment compared with the "mate" treatment (Supplemental Fig. S5). Clearly, being a female is not enough to elicit prosocial behavior in males, as the mere presence of the established mate caused antisocial choices by the subjects. As the mate and the new female did not see each other (Fig. 1), the subjects could not respond to any behavior of their mate and hence decided for themselves to make antisocial choices.

More generally, a key result of the present study was that male subjects are not just indifferent toward potential male or female competitors but actively make antisocial choices that prevent these competitors from obtaining food. This contrasts with results on primates, where subjects are typically only indifferent toward unfamiliar individuals[1]. Given that unfamiliar conspecifics are competitors, antisocial choices are what should be expected based on adaptationist thinking. The current results show parallels with studies on decision rules for social learning. In some social learning experiments, subjects should focus on outcomes, and/or they can choose to learn from different tutors who show alternative options. In such experiments, nine-spined sticklebacks and juvenile cleaner fish were found to focus on finding the best outcomes, i.e., they used so-called payoff-based social learning rules[35,36]. By contrast, primates generally prefer to learn from specific individuals even if alternative tutors show better solutions[37]. Potentially, the convict cichlids prefer antisocial choices because the intensity of intraspecific competition is stronger than it is in primates: conspecifics other than the partner of the male convict cichlid are always a threat at spawning sites[27,30]. Nevertheless, as the antisocial choice is cost-free in the PCT paradigm, it should also be the best option for primates if the potential recipient is a stranger.

**What are the implications for PCT in convict cichlids?** Hence, our results for the PCT performed by cichlid fish show the first evidence that fish may have both prosocial and antisocial motivations toward conspecifics. Because subjects could eat foods regardless of prosocial or antisocial choice, their prosocial and antisocial motivations are likely derived from seeing the recipient consume food at the proximate level[1]. The present study supports that highly advanced cognitive processes that would warrant a large endotherm forebrain are apparently not crucial for the expression of proactive prosociality. In addition, the study supports the hypothesis that at least in vertebrates, it is the social organization of a species, most notably, the interdependence between partners in raising offspring together, which causes the evolution of other-regarding preferences[4]. In our study species, the interdependence between the male subjects and their female partner is particularly high because of their joint reproduction. Such direct interdependence for reproduction compares with dominant male capuchin monkeys caring for the female group members. Conversely, in most primate constellations tested so far, the interdependence between familiar partners is linked to increased survival (female–female or breeder–helper). Testing group-living fish species would hence be the next step to investigate to what extent fishes may be capable to express proactive prosociality in a wider social network. In line with the ecological approach to cognition[38,39], fishes have already been shown to engage in transitive inferences[40,41], generalized rule learning[42],

referential gestures[43], and mirror self-recognition[44]. Our study also revealed that teleost fishes provide a model system for testing the triadic relationship between socio-ecological conditions, the expression of proactive prosociality, and underlying cognitive processes.

## Methods

**Subjects and fish husbandry**. The male and female adult convict cichlids, *A. nigrofasciata*, were obtained from ornamental fish traders. The fish were kept separately in three stock tanks (182, 364, and 97 L) at Osaka City University (Osaka, Japan). Furthermore, we kept the fish density and sex ratio constant between stock tanks, and the water temperature was maintained between 25.0°C and 27.0°C and the pH between 5.0 and 7.5. The fish were fed artificial food, and sexes were also readily distinguished in the adult state because of dimorphism in body size[27]. When a male and female showed a pair-specific display[45] and cared for the offspring in a stock tank, we regarded them as a breeding pair. To ensure homogeneity in a kin relationship between the male subject and the presented fish, the diverse family lines of adult males and females were present in each aquarium. In addition, we used fish unrelated to the subject fish from another aquarium, which was visually and olfactory isolated, in the unfamiliar fish experiments. All subject fish were males that had reproduced at least once. In total, we used 31 breeding pairs in this study (males, mean ± SD = 63.1 ± 2.3 mm SL, female, 52.7 ± 3.0 mm SL): we had 25 male subjects and their female partners and another six pairs as potential recipients in the "rival male" treatment, the "new female alone" treatment, and the "new female with subject's mate present" treatment.

**Experimental setup and training**. Two identical-size glass tanks ($36 \times 20 \times 20$ cm³, experimental and presentation tanks) were placed adjacent to one another so that fish could see into the neighboring tank (Fig. 1). The experimental tank contained two small transparent compartments ($10 \times 10 \times 20$ cm³) with doors, in which food was provided to the male subjects.

Phase 1—The male subjects were acclimated to the experimental tank, also learning to feed inside the compartments. During this phase, the female mate was also present in an attempt to keep the stress levels of the subjects as low as possible until the choice experiment (Fig. 1A). The doors and partition between the compartments were removed so that the fish could enter the compartments freely from the outside. We fed the fish with artificial food thrice per day. As soon as the male subjects ate all of the food provided in a day, phase 1 ended (usually ~1 week after starting).

Phase 2—The male subjects learned to take the food from the compartments after door opening. The movable doors were installed on each compartment, and the fish could enter the compartments only when the experimenter opened the transparent doors simultaneously by pulling a rope that was attached to both doors. We provided food thrice per day. Food was first dropped, and both doors were opened simultaneously 5 min later. This phase, which lasted ~10 days, ended when the male subjects waited consistently in front of the doors as soon as food had been dropped by the experimenter. During this phase, the female partners were still present and ate part of the food.

**Choice experiments**. Compared with the training, a few changes were made during experiments. First, a partition between the two feeding compartments was installed, and the doors to the two feeding compartments were marked with diverse shapes (quadrilateral, triangle, and round) and colors (blue, red, yellow, orange, and green) with the goal to enhance the ability of the subjects to learn to associate their choices with consequences for the recipients. One mark corresponded to a prosocial choice and the other mark to an antisocial choice. The shape–color combinations were fixed for one subject but randomly changed across treatments. If the male ate food in the prosocial compartment, then the same amount of food he was given by the experimenter into the center of the neighboring tank just after the male had finished eating. Finally, the mate was removed from the tank of the male subjects for the entire duration of the experiments. We tested the compartment choices of the males in five different conditions, defined by the identity of individuals present in the neighboring tank: "mate" (the subject's established female mate), "rival male," "new female alone," "new female with subject's mate present," and "control" (neighboring tank empty). The unfamiliar fish were taken from stock tanks different from that of the male subject, and the presented fish were all unrelated to the male subject.

Each treatment comprised nine trials per day over 10 consecutive days, yielding a maximum of 90 trials per treatment. Sessions of three trials were performed thrice per day at 9:00, 13:00, and 16:00. In some cases, subjects would not choose a compartment on every trial. Such missing values were not replaced by additional trials. Furthermore, if a male chose less than thrice in a day, then the data for that day were discarded. Finally, if a subject showed a strong side bias during the first 3 days, choosing one side (>80% of trials), then it was discarded from the experiment, and no further trials were conducted. In total, we had sample sizes of 12 (mate and control) and 10 (other three treatments) subjects for each of the five treatments.

In between distinct treatments, the males spent almost a month in a separate tank with their mate to ensure that the males were properly bonded before starting with the next treatment. During treatments that did not involve the mate as the potential recipient, the female remained in this tank, out of view.

**Behavioral observations beyond the subject's choice of compartment**. We recorded the behavioral interactions between the subject and recipients using a video camera (HDR-CX390; Sony, Tokyo, Japan) for later analyses. To determine whether the male subject monitored the consequences of his choices, we analyzed male orientation in detail and all prosocial choices with the mate as the recipient. We focused on the moment the subject had eaten its own food, which coincided with the recipient eating her food. The body orientation of the males when the recipient ate was classified into five categories, from directly facing the recipient (receiving a score of four) to facing in the opposite direction (receiving a score of 0). We knew that this measure is potentially not informative because fishes have a large field of vision. As shown in Supplementary Movie 1, the males sometimes quickly turned their body toward the recipient when the latter ate the food. As we considered this turning behavior as the strongest indicator of attention, we used its occurrence within an experimental day (1 = subject showed the behavior, 0 = subject did not show the behavior) to test whether attention varied as a function of the day. The hypothesis to be tested was that turning should occur more frequently during the first half of experiments, i.e., when males still needed to learn about the consequences of their actions (see Fig. 2). The body orientation and turning data were collected by an observer who was blind to the treatment and hypotheses. In addition, we had to exclude 376 data points from the sequential analysis, either because we could not obtain the correct positions of both fishes or because the recording failed. Thus, we analyzed 704 trials in this analysis.

A potential explanation for prosocial behavior could be that the spatial position of the potential recipient could affect the choices made by the subject fish[46]. If mates preferentially wait in front of the compartment that represents the prosocial choice, then males seeking proximity to their mates would act prosocially without intending to do so. On the basis of the video-recordings, we thus recorded the position of the mate at the moment the male made his choice and entered a compartment. We also distinguished three possible positions: in front of the prosocial compartment, in front of the antisocial compartment, and away from the compartments (Fig. 1c). The exact location of the head was applied if the entire body was between two positions. We quantified descriptively how much time the mates spent close the compartments, and we also tested whether they spent significantly more time in front of either compartment in each trial. Moreover, we had to exclude 39 data from the analysis as we either did not obtain the correct position of the mate from the video or the recording failed. Hence, we analyzed 1041 trials in this analysis.

**Statistical analyses**. All statistical analyses were performed using R software (v. 3.1.1; R Development Core Team, Vienna, Austria)[47]. As can be seen in Supplementary Table S6, the order of treatments was not properly counterbalanced for two reasons. This is because we initially focused on the comparison between the treatments "mate" and "control" and added the other three treatments only during the ongoing experiment. Second, we did not subject all subjects to all treatments for various reasons, meaning that subjects could have been tested in any possible number of treatments (between one and five). Consequently, we decided to run two statistical analyses on the basis of the advice of a statistical expert who was blind to our hypotheses and only concerned with data quality. In the first model, we compared only the treatments "mate" and "control." The order of presentation of these two treatments as either the first or the second half was almost perfectly counterbalanced. We hence utilized these data for a GLMM[48] with treatment and day as main factors, with individual as a random factor. On the basis of the results of this model, we found that subjects would develop a stable preference after 5 days of testing. We therefore only used days 6–10 for the full model, and this full model included all treatments and testing order as main effects (while day could be ignored), with individuals as random effects. As few individuals had been subjected to more than three treatments, we classified order into three groups (1, 2, and 3+). As the model yielded a significant interaction between treatment and order, we presented the model predictions for each order group separately.

We were mainly interested in whether males ever prefer the prosocial or the antisocial choice significantly above chance levels as a function of treatment. Thus, in the final analysis, we combined the values for the three separate order results into a single test against the random choice expectation of 50%.

We ran another two analyses, focusing on the treatment that involved the mate, to test whether recipients may have solicited food delivery and whether males tracked the consequences of the action. The body orientation index of the males was analyzed using a CLMM. The probability that the subjects performed a quick turn toward the feeding mate was tested as a function of the day based on a binomial GLMM. In these models, we fitted the early versus later parts of the experiment (1–5 days versus 6–10 days) as a binary explanatory term based on the analysis for the mate experiment (Fig. 2), with individual identity as a random factor.

To test whether the time that the mate spent in front of the prosocial or antisocial compartments affected the rate of prosocial choice by male subjects, we fitted the rate of prosocial choice as a response term of a binomial GLMM, with

experiment identity as a random factor. The percentage of the time female mates spent in front of the prosocial compartment and the antisocial compartment were explanatory terms, and experimental day (i.e., 1st to 10th) was fitted as a covariate (Supplemental Table S3). Similar to this model, we tested whether the position of the presented female mate, when the subject chose prosocial or antisocial compartments, affected the prosocial choice. We fitted the position of the presented female when the subject chose prosocial or antisocial compartments (i.e., within 10 cm in front of prosocial or antisocial compartment or far from compartments, Fig. 1c) as an explanatory term in this model (Supplemental Table S4).

**Ethics statement**. All the experimental protocols were approved by the Animal Care and Use Committees at Osaka City University for Advanced Studies and adhered to the ASAB/ABS guidelines for the treatment of animals in behavioral research.

**Reporting summary**. Further information on research design is available in the Nature Research Reporting Summary linked to this article.

## Data availability
The behavioral data that support the findings of this study are available in Dryad (https://doi.org/10.5061/dryad.k3j9kd565)[49].

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

## Acknowledgements

We thank the members of the Animal Sociology Laboratory of Osaka City University and the Kutsukake Research Group of Graduate University for Advanced Studies for their helpful comments. We also thank Ms. Yuka Shimamura of the Animal Sociology Laboratory of Osaka City University for her cooperation in blind behavioral observation. Furthermore, we thank Dr. Radu Slobodeanu who gave precious statistical advice. This study was supported financially by KAKENHI (nos. H17J11490, N19K23765, N19KK0189, and 20J01170 to S.S.).

## Author contributions

S.S., M.S., S.I., and M.K. designed the study. S.S., M.S., and S.I. collected the data. S.S., R.B., and S.A. conducted the analyses. S.S., R.B., S.S., H.T., and M.K. wrote the manuscript with input from all the authors.

## Competing interests

The authors declare no competing interests.
