## [Peer Review File · Nature Communications]

Reviewer #1 (Remarks to the Author):

Review of: Other-regarding preference in fish: Evidence from a Prosocial choice task in a monogamous cichlid
Nature Communications

The manuscript entitled: Other-regarding preference in fish: Evidence from a prosocial choice task in a monogamous cichlid presents evidence of selective, proactive prosocial preferences through a series of experiments in male cichlid fish. This experiment represents an important test of the evolutionary hypothesis that social interdependence underlies performance on the PCT. Running this experiment on a taxon so distantly related from those previously studied represents an important advance in our understanding of how other regarding preferences have evolved.

As I felt this study was generally well designed, I only have one major comment:

I am a bit concerned about highly variable experimental order. While I acknowledge that the authors consulted a statistician about the best way to deal with this in the analysis, I still have some concerns, particularly as the results are described as followed: "taken together, these results indicate that males apparently understood the consequences...". But it is unclear to me if that can be concluded since so many of the individuals did not participate in the full suite of tests. For example, was the first analysis limited to only males who participated in both an experimental and a control condition? If not, do the results hold if limited to only those males? While I realize some PCT tasks (such as the capuchin one this study seems to be heavily based off of) tend to only compare behavior to chance and treat each condition separately, a truer measure of "understanding the consequences of their prosocial and antisocial choices" (line 98) would be to compare each males behavior in the mate vs. control. In other words, the males should flexibly alter their behavior and adjust to each condition if they truly understand what the different choices are. You should see a significant difference between the number of prosocial choices in the mate condition vs. the control condition, not each one compared to chance. Given that there was an effect of order on the later test conditions, this suggests that the males might not be responding as flexibly to the changing conditions or that their understanding of the options is being shaped by their previous experiences.

To rectify this situation, I would like to see the first analysis (mate vs. control) limited to only the males who completed both as their first two conditions (I1, I5, S1, S5, S6). If there is enough power, I would like to see their choices compared across these two conditions. This would be the strongest test of their understanding. If this is not possible, it should be included as a limitation. And again, I acknowledge this has not consistently been done in PCT studies to date (particularly those done in primates) but it should have been done, and it would do a disservice to this area of the literature to continue omitting a key comparison needed to assess understanding.

I also have several minor comments, which I have outlined by section and line number below.

Introduction:

Ln 17: Is "active" prosociality the same thing as proactive? I suspect that it is, but wanted to clarify. If so, the paper might benefit from consistent wording throughout as the word active also appears in line 62.

Ln 19-26: While the consistency of results among cooperatively breeding primates is impressive, much of the current study relies on earlier work in capuchins, which consistently defy these hypotheses. It would be good to situate the capuchin work within the broader discussion of social interdependence, since it is so important to the current study.

Ln 51: Would it be possible to create a new paragraph to transition to the current study? I was initially confused by the transition from basic information about cichlids to the set up of the current study.

Ln 65-66: There is a bit of inconsistency in the language used to describe selfish choices. Here it is described as "active avoidance of prosocial behavior" later it seems to switch back and forth between selfish (which is the term frequently used to describe that compartment, as well as the choice on line 447) and antisocial (lines 98, 125, 143, etc.). If these are all referring to the same thing, it might be helpful to the reader to call them one thing consistently. If they are different, can you please clarify what each term means in the text somewhere?

Ln 64-69: I seem to be missing the hypothesis for the condition of the new female in the presence of the mate? I see information for all of the other conditions but not that one. In the discussion (lines 135-137) there is some information on how the presence of the mate might impact treatment of stranger females. This might be useful to move to the introduction as it provides the basis for hypothesizing that a new female alone might be treated differently than a new female with a mate present.

Results and discussion

Ln 82: I am not sure what is meant by "behaviorally examined"? Is this referring to the attention towards the partner? In other words, the monkeys paid attention to the impact of the outcome on their partner? Can you clarify this term?

Lines 102-114: While I agree that an alternative hypothesis is that the prosocial choice delivers more food overall, it's unclear to me how manipulating the social relationship variable answers that question. To me, the appropriate control for that alternative would be to create a situation where there is an empty compartment where food is delivered out of reach of the mate during the selfish choice. That way the same amount of food is delivered each time, what differs is whether the mate can actually eat it. I believe the social manipulations in the capuchin studies you described were based on the premise that empathy exists on a gradient, so if empathy underlies other-regarding preferences, you should see the strongest response to kin, then non-kin group-mates, then strangers. While I think much of the literature strongly supports the idea that social relationship plays a role, and indeed, it should if the evolutionary hypothesis is that other-regarding preferences arise from social interdependence, it just seems to me there is a disconnect between the alternative you suggest in lines 102-105 and the follow up experiments you did varying the social relationship with the partner.

Lines 104 and 110: I think you have the word "compartment" where you mean "case". Also on line 226.

Methods:

Ln 180: For the unfamiliar fish, were they housed out of visual contact? (I am also presuming out of olfactory contact because they were in separate tanks, but it would be good to explicitly state that if that's the case).

Ln 206: While I understand why the female was kept in the tank in Phase 1, I am confused why she was still present at this point. Why was she not removed as part of the initial acclimation? I am concerned that her presence during this part and the fact that they ate together, may have conditioned the male to expect that they should eat together in the tank, which could then lead to increased prosocial choices during the test phase. This seems unlikely to me given how consistently the males seem to switch choices as a result of social partner, and the fact that half of the males had the control condition (where they were alone first). But as this study has an otherwise fairly tidy design where alternatives and confounds were carefully avoided or controlled for, this detail is concerning to me, and it would be nice to see justification for the female's presence in phase 2.

Lines 238: Was inter-rater reliability done for the behavioral observations? If I am reading table S4 correctly, different raters scored different fish, so a measure of inter-rater reliability should be used to make sure they agree with each other.

Line 284: When you say “the presentation as either first or second” does that mean fish who were exposed to the control beyond the second experiment (fish S3 and S10 according to table S4) were excluded from this analysis?

Lines 290-293 and Figure 3: I am not sure why I am not understanding this, but I am very confused by the order1, order 2 and order 3+. First, I think it would be good to reference table S4 somewhere in this section of text. Second, is order 1 referring to the first experiment? So is the first panel on figure 3, the fish who did that condition first? If so, maybe renaming the panels/orders as Experiment 1, Experiment 2, Experiment 3+ might help readers like me who are having difficulty understanding the different orders. I suspect that is not the case, however, because I see in table S4 that no fish had the mate or the new female with mate present in experiments 3-5. But there are data represented on panel 3 of Figure 3. Can you please clarify this?

I hope these suggestions are useful to the authors and taken in the spirit in which they are meant.

Reviewer #2 (Remarks to the Author):

This study is exploring the presence of active prosociality in the monogamous convict cichlid fish, adapting a paradigm extensively used in primates. Active prosociality is the subject of a large number of studies on primates, and it has been suggested that it may be more likely in species with strong social interdependence, given that these species would show decreased competition between partners whose fitness is linked.

This study is thoughtfully and carefully planned and well executed; the authors control for many different factors, such as the effects that different stimulus individuals will have on the expression of prosociality. The statistical approaches followed are completely appropriate, and issues such as whether the focal individual pays attention to the outcomes of its actions are rightfully and appropriately taken into account. The writing is very clear for the most part.

My main concern regarding this manuscript is a confusion of terms regarding prosociality, antisociality, and interdependence between social partners. For example, the authors report that focal individuals made prosocial choices when paired with their female mate, or with an unfamiliar female in the absence of her partner; they also report antisocial choices when males were paired with other males, or females in the presence of their partner. The authors suggest that prosociality is the most likely explanation and exclude kin selection as a possible mechanism at play here (line 131). I believe that there is another likely mechanism that doesn't get thoroughly discussed: as the authors say, male fitness in this species is interdependent with the fitness of female partners, as individuals form stable pairs for biparental care of the brood (lines 132-133). Therefore, when males make the 'prosocial' choice when paired with their mates, or other females that do not have a male mate in sight, they are likely increasing their own reproductive success, by increasing their mate's fitness (via food). The same happens in the unfamiliar female treatment: as the focal male has been isolated from their female mate for the duration of the experiment (10 days), it is likely that they perceive this unfamiliar female as a possible mate – in which case, the prosocial choice also increases the likelihood of reproductive success. This suggests that an alternative explanation to that of overall prosociality is very likely, i.e. that males base their choices not on helping others, but on increasing their own reproductive success. The studies on primates that the authors cite (e.g de Waal et al., 2008; Burkart et al., 2007; Horner et al., 2011) look at decisions between unfamiliar and/or bonded familiar conspecifics; however, primate societies differ a lot to the social system of *Amaltilania nigrofasciata*, in that the food intake of the social partner does not directly affect the reproductive success of the focal individual, as is the case in this study system. In fact, the authors discuss interdependence in their introduction and its link to prosociality; however, that link is not made very clear. The Prosocial

Choice Task itself forces a choice between a selfish option that rewards only the actor and a prosocial option that rewards both the actor and others; however, in the case of a male fish paired with his mate, contrary to the standard PCT in primates, the prosocial choice offers the actor a higher reward than the selfish choice, given that a mate with higher fitness will provide better parental care, thus affecting the actor's reproductive success, and suggesting that prosociality in this context would affect the overall reproductive success.

I believe that the authors should clearly discuss that there are probably more than one mechanisms at play here, and that the emergence of prosociality is likely linked to the interdependence of factors; it is therefore extremely difficult to disentangle prosociality from increasing one's own reproductive fitness in this context and study system. The use of clear definitions throughout the manuscript would help the reader follow the rationale of this experimental approach, and the clearer use of terms such as 'prosociality' and 'antisociality' would help discern between prosociality, cooperation and interdependence, and aid the interpretation of the results.

Other comments:

Line 50: when the authors refer to 5 reproductive cycles, it would be useful to clarify the timescale of 1 reproductive cycle.

Line 104: it's unclear what the authors refer to as a 'compartment' – it may be more useful to refer to experimental trials instead?

Lines 144-146: this sentence is unclear – consider rewriting

Lines 152-155: this sentence is unclear – consider rewriting and expanding on the kind of decision rules in fishes and primates, and how they differ

Lines 211-217: I am not entirely sure what the point of colour on the compartment doors is – it would be nice to see that clarified. Also, was there any bias towards a specific side/colour?

Line 229: the authors state that if a male chose less than 3 times in a day, the data were discarded – was the data discarded for that day only, or was that individual removed altogether from the dataset?

Line 242: It is very good that the authors have examined whether the focal individual was paying attention to whether the stimulus fish received food; it is, however, unclear whether that only happened in trials where the stimulus fish was the female mate or across all trials. This would be an important thing to mention, as the order of the trials differed between individuals.

Line 259: substitute 'potential' for 'potentially'

Reviewer #3 (Remarks to the Author):

1. Key results: Please summarise what you consider to be the outstanding features of the work.

The authors report novel findings that male convict cichlid fish make active choices to be prosocial or antisocial depending on the sex and novelty of the potential recipient. They have also generated an appropriate a prosocial choice task suitable for application to fish which has previously been limited to land-borne vertebrates. The manuscript itself is concisely and clearly written and in general, follows a logical format.

2. Validity: Does the manuscript have flaws which should prohibit its publication? If so, please provide details.

Currently there are no outstanding flaws that would prohibit the publication of this paper. There are also no special ethical concerns arising from the use of animals, with the authors clearly taking care to habituate individuals to isolation from their mate. However, as reported in point 4. there are key details missing which may point to a flaw and statistical method would need to be reviewed by an independent expert (point 11).

3. Originality and significance: If the conclusions are not original, please provide relevant references. On a more subjective note, do you feel that the results presented are of immediate interest to many people in your own discipline, and/or to people from several disciplines?

As far as I am aware, these conclusions are original. These findings will be of immediate interest to those researching models of social decision-making across clades and other research using this particular species of fish (widely used in reproductive experiments).

4. Data & methodology: Please comment on the validity of the approach, quality of the data and quality of presentation. Please note that we expect our reviewers to review all data, including any extended data and supplementary information. Is the reporting of data and methodology sufficiently detailed and transparent to enable reproducing the results?

Overall the data are clearly presented, with the nice use of diagrams to illustrate the experimental design.

However, currently there is not enough detail to allow reproduction of experiments and does not show that the authors have used previous work on this species to inform decisions on design. The manuscript would greatly benefit from the clarification and inclusion of key methodological details and justification for these as noted below:

Line 173. Were all fish provided from the same ornamental traders, and of the same age? There could be baseline genetic differences between individuals which would need to be balanced across treatments or used as a blocking effect in the final statistical model.

Was any effort made to balance the size or age of fish between treatments/pairs/original stock tanks? Stocking density differed between the three holding tanks which could potentially have an impact on subsequent fish behaviour. Were efforts made to balance across the tanks so that there was equal opportunity to form a pair bond?

What happened to the pairs after pair bonding had occurred? Were they kept in separate tanks with ideal spawning conditions (e.g. clay pot as spawning substrate, ideal temperature?). Further, how was a successful pair bond defined? Itzkowitz and Bockelman (2008) (<https://doi.org/10.1163/156853908783402939>) provide examples of specific criteria used based on display of two courtship activities. This could influence the strength of the decision for prosociality in the subsequent test.

It may help to clarify that this species shows clear sexual dimorphism so that the male subject are capable of distinguishing between individuals or at least different sexes.

Line 187. Was the same experimental tank used for all subjects? Was the water changed between subjects to prevent a potential influence of circulating hormones released by previous subjects?

Line 197. Using the time taken to finish all the food or habituate could be an interesting additional variable to indicate the personality or anxiety levels of each fish.

Line 204. Why was this determined as the end point for phase 2? I assume it is because they have learned to go to the doors to get the food.

Line 212. Were there differences between colour and shape (as shown in Fig 1.)? Which colours were used and have previous experiments shown that fish can differentiate between them or show any preference for a certain combination (e.g. certain colours could be innately associated with a preferred food source).

Line 225. Were all trials conducted at the same time of day? I am unsure if these fish show any circadian rhythms!

Please clarify what is meant by a 'trial' and were there any gaps between trials? Was there any evidence of satiation by the end of the 9th trial for each day? Why was this number of trials chosen? Was it based on previous literature/a statistical power test/time limitations?

Line 238. Are any of these behavioural measurements based on previous literature i.e. other fish choice tasks/showing mate preference?

Another potential behavioural measurement could be the time the male subject took to make the decision, which could indicate a strength of decision making in addition to the binomial prosocial/antisocial.

Line 229. A potential way of investigating side bias could have been to swap the location of colours randomly between trials for the same subject. This would provide a definite answer to whether the fish was showing side bias rather than making a consistent choice. Further, were the biased fish always showing preference for the same side? Could this be explained by an environmental effect?

Was the collection of behavioural observations conducted by the same person? If so, were any efforts made to check intra-observer reliability?

Line 309. For clarity it would help to state why this period was chosen. Is this based on the previous model's results or on the previously stated hypothesis?

Supplemental data;

S5 – I don't think it is necessary or appropriate to include a direct screenshot of a statistical output.

Video – this is a good idea however it could be improved. The current video is short and of poor quality due to reflection and has no indicator of what it is showing. A longer video with subtitles/commentary would be more influential.

5. Appropriate use of statistics and treatment of uncertainties: All error bars should be defined in the corresponding figure legends; please comment if that's not the case. Please include in your report a specific comment on the appropriateness of any statistical tests, and the accuracy of the description of any error bars and probability values.

Error bars are present in the appropriate figures and defined – what statistical method used to generate these intervals is not defined. In Figure 2. the 95% CIs appear to be the same across all trials for the antisocial choice but differ between trials in the prosocial choice – is this correct?

Figures 2 and 3 show no information on statistical significance or p values.

Figure legends are sparse – particularly Figures 2, 3 and 4, as well as the supplemental data. More information is needed e.g. What means were used and from which model?

6. Conclusions: Do you find that the conclusions and data interpretation are robust, valid and reliable?

Data interpretation is valid, with results and discussion both following the same conclusions.

7. Suggested improvements: Please list additional experiments or data that could help strengthening the work in a revision.

The suggestions below are not necessary for this experiment but are thoughts to be considered in further studies.

Size appears to be an important factor in mate choice for these species, so I am surprised there is no mention of its consideration in this study, with both sexes preferring large mates. I would expect size to be used to balance between treatments and could also be used as covariate for the strength of the other-regarding preference. Size is likely to be an influential factor if these experiments were repeated the females as the subject, considering they will swap mates for a larger mate (see Gagliardi-Seeley, 2009 - 10.1007/s10164-008-0111-2).

Reproductive status of the new female or current mate could be another factor to consider and have an influence on the male's choice, particularly when both females were present. This could easily be quantified through used of the female's 3 distinct colour phases, which correspond to their reproductive status (Robart and Sinervo, 2018 - <https://doi.org/10.1093/beheco/ary028>).

8. References: Does this manuscript reference previous literature appropriately? If not, what references should be included or excluded?

I feel it would be advisable for the authors to review other papers using this particularly animal model, such as those referenced in the comments above. This would help to fill in some of the information I feel is missing (again as noted above) to establish whether these factors have been overlooked during experimental design (e.g. reproductive status).

9. Clarity and context: Is the abstract clear, accessible? Are abstract, introduction and conclusions appropriate?

The abstract is perhaps too concise in terms of information. Accessibility and clarity could be improved through defining what is meant by 'proactive prosociality' in the first sentence, what is briefly involved in the task and what is meant by a 'behaving prosocially', i.e. eating out of side that meant the other fish also got food.

The introduction follows a clear and logical flow of previous studies and arguments and the introduction of the aim of the study. Further improvement could be made through further justification of the logic and reference ideas behind key experimental questions as raised below;

Line 55. Please clarify why only males were used as the test subject.

Line 62. It would help to repeat here what the control condition was.

Line 64. Provide a brief explanation of how this hypothesis was tested

Line 64-69. Are there any references from studies in other species to support the design of your test and logic of your hypotheses? The 'foraging rate' justification for adding these extra treatments is provided in the discussion but not here?

The overall conclusion is appropriate and clearly placed within wider hypotheses and discussed in the context of previous literature of the interdependence between the nature of reproductive relationships (e.g. the presence of parental care allocation) and the presence of prosociality.

10. Please indicate any particular part of the manuscript, data, or analyses that you feel is outside the scope of your expertise, or that you were unable to assess fully.

The final statistical model reported would benefit from revision by a statistical expert. The authors have made a clear effort to take account of potential order effects and the unbalanced allocation of treatments. I feel I am not qualified to ascertain the appropriateness of this model and whether there are sufficient or valid replicates.

Tayla J. Hammond

Reviewer #4 (Remarks to the Author):

This simple, but elegant experimental study shows how proactive (but cost-free) prosociality is linked to the presence of strong social bonds and parental care relationships, and thus provides strong support for the interdependence hypothesis for proactive prosociality. The study controls for the possibility that the prosocial acts are reactive, and that the subjects simply liked seeing more food around (by adding a rival as possible recipient). Overall, this is a very convincing and well-designed study. I can find nothing major to object or question, and it would make a valuable addition to the literature. I must admit that I am not a fish expert, so cannot judge whether the natural history information on this species is presented correctly (but it seems to me it is).

A second attractive feature of the paper is that the results suggest an awareness of the risk of partner loss when alternative partners are presented in view of the current long-term partner. This result fits neatly into the wave of recent studies showing so far unexpected cognitive abilities in fishes.

I cannot find much to object to, but hope the authors can reassure me that the exclusion of individuals with a persistent side bias (P9, l229) cannot have affected the results that were obtained.

Detailed questions or comments:

P2, l34 It is fair to say these cognitive abilities are necessary for the findings reported in this paper, so merely saying that they have been shown in some fishes is, strictly speaking, not enough: they must be demonstrated to be present in the species that was tested. This requirement should not sink the paper, but some mention of this assumption is required.

P3, l38-43 These lines do not belong in the introduction, but in the discussion, since they have no bearing on the predictions.

P3, l55 It would be helpful to explain why only males were tested.

P4, l82 The monkeys behaviorally examined the other-regarding preference. What does that mean? If they mean whether donors checked whether recipients took the food or were interested, in other words whether they intentionally produced food for their mates, this has been studied rigorously more recently (see Burkart & van Schaik 2020, Anim Cogn).

P5, l109-10 I do not understand "which is indeed often the compartment" Please clarify.

P5, l120 I did not see an attempt to explain the minor differences in the three treatment order groups as a function of differential experience or the order in which the experience was gained. Perhaps this could be added to the supplementary information.

P6, l137+ One would expect an effect of the time elapsed since the male was separated from his mate on the tendency to provision a strange female. This could easily be checked in the data.

P9, l229 I do not understand how fishes can have a side bias when during the whole experiment the food compartment is always one and the same. This is the only aspect of the study that I find worrisome, so please explain.

P10,259 potential rather than potentially, unless an adjective is missing after potentially.

P12, l312+ Why are these analyses done on the aggregate data rather than on a per-trial basis?

Finally, in various places, the authors refer to 'active' prosociality, which blurs the distinction between proactive and reactive. So, please either define it or use the terms proactive and reactive.

Referee#1

The manuscript entitled: Other-regarding preference in fish: Evidence from a prosocial choice task in a monogamous cichlid presents evidence of selective, proactive prosocial preferences through a series of experiments in male cichlid fish. This experiment represents an important test of the evolutionary hypothesis that social interdependence underlies performance on the PCT. Running this experiment on a taxon so distantly related from those previously studied represents an important advance in our understanding of how other regarding preferences have evolved. As I felt this study was generally well designed, I only have one major comment:

I am a bit concerned about highly variable experimental order. While I acknowledge that the authors consulted a statistician about the best way to deal with this in the analysis, I still have some concerns, particularly as the results are described as followed: “taken together, these results indicate that males apparently understood the consequences…” . But it is unclear to me if that can be concluded since so many of the individuals did not participate in the full suite of tests. For example, was the first analysis limited to only males who participated in both an experimental and a control condition? If not, do the results hold if limited to only those males? While I realize some PCT tasks (such as the capuchin one this study seems to be heavily based off of) tend to only compare behavior to chance and treat each condition separately, a truer measure of “understanding the consequences of their prosocial and antisocial choices” (line 98) would be to compare each males behavior in the mate vs. control. In other words, the males should flexibly alter their behavior and adjust to each condition if they truly understand what the different choices are. You should see a significant difference between the number of prosocial choices in the mate condition vs. the control condition, not each one compared to chance. Given that there was an effect of order on the later test conditions, this suggests that the males might not be responding as flexibly to the changing conditions or that their understanding of the options is being shaped by their previous experiences.

To rectify this situation, I would like to see the first analysis (mate vs. control) limited to only the males who completed both as their first two conditions (I1, I5, S1, S5, S6). If there is enough power, I would like to see their choices compared across these two conditions. This would be the strongest test of their understanding. If this is not possible, it should be included as a limitation. And again, I acknowledge this has not consistently been done in PCT studies to date (particularly those done in primates) but it should have been done, and it would do a disservice to this area

of the literature to continue omitting a key comparison needed to assess understanding.

Response to major comment

We are grateful to Reviewer#1 for the comments that offered to improve our manuscript. We hope the new version of our manuscript would be more sophisticated. Major changes we have made are addition of analysis based on your comments. In this part, we additionally have added new analysis limited to only the male who completed both of mate and control experiments and gained similar results we made [L87-89]. Because information is overlapping with Fig.2, figure of additionally analysis are shown in supplemental data S1.

Response to minor comments

Comment 1

Is “active” prosociality the same thing as proactive? I suspect that it is, but wanted to clarify. If so, the paper might benefit from consistent wording throughout as the word active also appears in line 62.

Response

Thank you for your comments. We have changed “active” to “proactive” and unified consistent wording throughout.

Comment 2

While the consistency of results among cooperatively breeding primates is impressive, much of the current study relies on earlier work in capuchins, which consistently defy these hypotheses. It would be good to situate the capuchin work within the broader discussion of social interdependence, since it is so important to the current study.

Response

Our study was inspired by capuchin work; therefore, we have changed some sentences. In L110-120, we have emphasized the perdition the effects of social relationship between subject and experimental partner as capuchin work. Additionally, in L174-176, we have supplemented the “conclusion” section with explanation of the relationship with “self-rewarding” .

Comments 3

Ln 65-66: There is a bit of inconsistency in the language used to describe selfish choices. Here it is described as “active avoidance of prosocial behavior” later

it seems to switch back and forth between selfish (which is the term frequently used to describe that compartment, as well as the choice on line 447) and antisocial (lines 98, 125, 143, etc.). If these are all referring to the same thing, it might be helpful to the reader to call them one thing consistently. If they are different, can you please clarify what each term means in the text somewhere?

Response

Thank you for your suggestion. We have rewritten to be more in line with your comments. Moreover, we have changed “selfish” to “antisocial” throughout the paper.

Comments 4

Ln 64-69: I seem to be missing the hypothesis for the condition of the new female in the presence of the mate? I see information for all of the other conditions but not that one. In the discussion (lines 135-137) there is some information on how the presence of the mate might impact treatment of stranger females. This might be useful to move to the introduction as it provides the basis for hypothesizing that a new female alone might be treated differently than a new female with a mate present.

Response

Thank you for your suggestion. We have added the prediction of “new female with presentation” experiment. [L74-77]

Comments 5

Ln 82: I am not sure what is meant by “behaviorally examined” ? Is this referring to the attention towards the partner? In other words, the monkeys paid attention to the impact of the outcome on their partner? Can you clarify this term?

Response

We removed “behaviorally” and hope that the deletion clarifies the points we attempted to make. [L91].

Comments 6

Lines 102-114: While I agree that an alternative hypothesis is that the prosocial choice delivers more food overall, it’s unclear to me how manipulating the social relationship variable answers that question. To me, the appropriate control for that alternative would be to create a situation where there is an empty compartment where food is delivered out of reach of the mate during the selfish choice. That way the same amount of food is delivered each time, what differs is whether the mate can actually eat it. I believe the social manipulations in the capuchin studies you

described were based on the premise that empathy exists on a gradient, so if empathy underlies other-regarding preferences, you should see the strongest response to kin, then non-kin group-mates, then strangers. While I think much of the literature strongly supports the idea that social relationship plays a role, and indeed, it should if the evolutionary hypothesis is that other-regarding preferences arise from social interdependence, it just seems to me there is a disconnect between the alternative you suggest in lines 102-105 and the follow up experiments you did varying the social relationship with the partner.

Response

Thank you for providing these insights. We agree with you. We have changed this section to establish a clearer focus to our evolutionary prediction and reflected this comment into [L110-120].

Comments 7

Ln 180: For the unfamiliar fish, were they housed out of visual contact? (I am also presuming out of olfactory contact because they were in separate tanks, but it would be good to explicitly state that if that's the case).

Response

Yes. Their house tank was completely separated each other. We have added sentence to explain this thing. [L209]

Comments 8

Ln 206: While I understand why the female was kept in the tank in Phase 1, I am confused why she was still present at this point. Why was she not removed as part of the initial acclimation? I am concerned that her presence during this part and the fact that they ate together, may have conditioned the male to expect that they should eat together in the tank, which could then lead to increased prosocial choices during the test phase. This seems unlikely to me given how consistently the males seem to switch choices as a result of social partner, and the fact that half of the males had the control condition (where they were alone first). But as this study has an otherwise fairly tidy design where alternatives and confounds were carefully avoided or controlled for, this detail is concerning to me, and it would be nice to see justification for the female's presence in phase 2.

Response

In our preliminary experiments, if female mate keeps in presentation tank longer than our experimental paradigm, female often spawned eggs nearby experiment tank. In this

care, we must stop our stop the experiments. Moreover, we thought their pair-bonding will be unstable even if we separate only few days. Therefore, we established and decided this experimental paradigm.

Comments 9

Lines 238: Was inter-rater reliability done for the behavioral observations? If I am reading table S4 correctly, different raters scored different fish, so a measure of inter-rater reliability should be used to make sure they agree with each other.

Response

I'm sorry it's hard to understand. Experiments were performed by three experimenters but behavioral observation was performed by only one observer under the blind condition. We have changed "observer" to "experimenter" in supplemental data S6.

Comments 10

Lines 290-293 and Figure 3: I am not sure why I am not understanding this, but I am very confused by the order1, order 2 and order 3+. First, I think it would be good to reference table S4 somewhere in this section of text. Second, is order 1 referring to the first experiment? So is the first panel on figure 3, the fish who did that condition first? If so, maybe renaming the panels/orders as Experiment 1, Experiment 2, Experiment 3+ might help readers like me who are having difficulty understanding the different orders. I suspect that is not the case, however, because I see in table S4 that no fish had the mate or the new female with mate present in experiments 3-5. But there are data represented on panel 3 of Figure 3. Can you please clarify this?

Response

We found mistake in the experimental order in supplemental table S4 comparing with real data set. This error has been corrected, and we have changed to "order 1-5" . Thank you for your suggestion.

Comment 11

Ln 51: Would it be possible to create a new paragraph to transition to the current study? I was initially confused by the transition from basic information about cichlids to the set up of the current study.

Response

We followed this comment.

Again, thank you for giving us the opportunity to strengthen our manuscript with your

valuable comments and queries.

Referee#2

This study is exploring the presence of active prosociality in the monogamous convict cichlid fish, adapting a paradigm extensively used in primates. Active prosociality is the subject of a large number of studies on primates, and it has been suggested that it may be more likely in species with strong social interdependence, given that these species would show decreased competition between partners whose fitness is linked.

This study is thoughtfully and carefully planned and well executed; the authors control for many different factors, such as the effects that different stimulus individuals will have on the expression of prosociality. The statistical approaches followed are completely appropriate, and issues such as whether the focal individual pays attention to the outcomes of its actions are rightfully and appropriately taken into account. The writing is very clear for the most part.

My main concern regarding this manuscript is a confusion of terms regarding prosociality, antisociality, and interdependence between social partners. For example, the authors report that focal individuals made prosocial choices when paired with their female mate, or with an unfamiliar female in the absence of her partner; they also report antisocial choices when males were paired with other males, or females in the presence of their partner. The authors suggest that prosociality is the most likely explanation and exclude kin selection as a possible mechanism at play here (line 131). I believe that there is another likely mechanism that doesn't get thoroughly discussed: as the authors say, male fitness in this species is interdependent with the fitness of female partners, as individuals form stable pairs for biparental care of the brood (lines 132-133). Therefore, when males make the 'prosocial' choice when paired with their mates, or other females that do not have a male mate in sight, they are likely increasing their own reproductive success, by increasing their mate's fitness (via food). The same happens in the unfamiliar female treatment: as the focal male has been isolated from their female mate for the duration of the experiment (10 days), it is likely that they perceive this unfamiliar female as a possible mate - in which case, the prosocial choice also increases the likelihood of reproductive success. This suggests that an alternative explanation to that of overall prosociality is very likely, i. e. that males base their choices not on helping others, but on increasing their own reproductive success. The studies on primates that the authors cite (e.g de Waal et al., 2008; Burkart et al., 2007; Horner et al., 2011) look at decisions between unfamiliar and/or bonded familiar conspecifics;

however, primate societies differ a lot to the social system of *Amaltilania nigrofasciata*, in that the food intake of the social partner does not directly affect the reproductive success of the focal individual, as is the case in this study system. In fact, the authors discuss interdependence in their introduction and its link to prosociality; however, that link is not made very clear. The Prosocial Choice Task itself forces a choice between a selfish option that rewards only the actor and a prosocial option that rewards both the actor and others; however, in the case of a male fish paired with his mate, contrary to the standard PCT in primates, the prosocial choice offers the actor a higher reward than the selfish choice, given that a mate with higher fitness will provide better parental care, thus affecting the actor's reproductive success, and suggesting that prosociality in this context would affect the overall reproductive success.

I believe that the authors should clearly discuss that there are probably more than one mechanisms at play here, and that the emergence of prosociality is likely linked to the interdependence of factors; it is therefore extremely difficult to disentangle prosociality from increasing one's own reproductive fitness in this context and study system. The use of clear definitions throughout the manuscript would help the reader follow the rationale of this experimental approach, and the clearer use of terms such as 'prosociality' and 'antisociality' would help discern between prosociality, cooperation and interdependence, and aid the interpretation of the results.

Response to major comment

We thank the referee for these thoughts. We have adjusted the ms at three places to be explicit about the very direct fitness link in our cichlids due to joint reproduction, as opposed to most primate studies where the benefits are linked to survival rather than reproduction [L137-138; 182-184; 184-187]. Only when male monkeys help their female harem members, the situation is directly comparable. Nevertheless, it is important to note that the interdependence hypothesis is always about direct fitness benefits. It is not clear to us whether survival and reproduction require distinct prosocial mechanisms. Moreover, A major change associated with your comments is that we have changed "antisocial" consistently throughout the text (following Silk et al. 2005).

Response to minor comments

Comment1

Line 50: when the authors refer to 5 reproductive cycles, it would be useful to clarify the timescale of 1 reproductive cycle.

Response

Thank you for your suggestion. We have added the describing of time scale of one reproductive cycle. [L41-54]

Comment2

Line 104: it's unclear what the authors refer to as a 'compartment' - it may be more useful to refer to experimental trials instead?

Response

I'm sorry it is our mistake but this section is now completely different with previous manuscript because we have changed based on referee#1's comment. Thank you for your suggestion.

Comment3

Lines 144-146: this sentence is unclear - consider rewriting.

Response

We have redrafted this section [L152-154] to establish a clearer focus.

Comment4

Lines 152-155: this sentence is unclear - consider rewriting and expanding on the kind of decision rules in fishes and primates, and how they differ

Response

We have expanded the text to better explain what we mean[L160-167].

Comment5

Lines 211-217: I am not entirely sure what the point of colour on the compartment doors is - it would be nice to see that clarified. Also, was there any bias towards a specific side/colour?

Response

Our focus of the colour on the compartment doors is supporting their associative learning of choices. That colour and their combinations was completely randomized. We have included new explanations of the colours. [L242-244]

Comment6

Line 229: the authors state that if a male chose less than 3 times in a day, the data

were discarded - was the data discarded for that day only, or was that individual removed altogether from the dataset?

Response

We discarded the data for only that day. We have revised the text [L260] to make clearly understandable.

Comment7

Line 242: It is very good that the authors have examined whether the focal individual was paying attention to whether the stimulus fish received food; it is, however, unclear whether that only happened in trials where the stimulus fish was the female mate or across all trials. This would be an important thing to mention, as the order of the trials differed between individuals.

Response

You have raised an important question. We also suffered whether that only happened in “mate” experiment or not. But we could not solve this question. For instance, in rival male experiment, subject always directed at rival male because it is their aggressive display. Therefore, we could not observe their behavior with the same criteria of mate experiment. We think that this question will be an important matter for future studies.

Comment8

Line 259: substitute ‘potential’ for ‘potentially’

Response

Thank you for your suggestion. We have changed it.

Again, thank you for giving us the opportunity to strengthen our manuscript with your valuable comments and queries.

Referee#3

1. Key results: Please summarise what you consider to be the outstanding features of the work.

The authors report novel findings that male convict cichlid fish make active choices to be prosocial or antisocial depending on the sex and novelty of the potential recipient. They have also generated an appropriate a prosocial choice task suitable for application to fish which has previously been limited to land-borne vertebrates. The manuscript itself is concisely and clearly written and in general, follows a logical format.

2. Validity: Does the manuscript have flaws which should prohibit its publication? If so, please provide details.

Currently there are no outstanding flaws that would prohibit the publication of this paper. There are also no special ethical concerns arising from the use of animals, with the authors clearly taking care to habituate individuals to isolation from their mate. However, as reported in point 4. there are key details missing which may point to a flaw and statistical method would need to be reviewed by an independent expert (point 11).

3. Originality and significance: If the conclusions are not original, please provide relevant references. On a more subjective note, do you feel that the results presented are of immediate interest to many people in your own discipline, and/or to people from several disciplines?

As far as I am aware, these conclusions are original. These findings will be of immediate interest to those researching models of social decision-making across clades and other research using this particular species of fish (widely used in reproductive experiments).

4. Data & methodology: Please comment on the validity of the approach, quality of the data and quality of presentation. Please note that we expect our reviewers to review all data, including any extended data and supplementary information. Is the reporting of data and methodology sufficiently detailed and transparent to enable reproducing the results?

Overall the data are clearly presented, with the nice use of diagrams to illustrate the experimental design.

However, currently there is not enough detail to allow reproduction of experiments and does not show that the authors have used previous work on this species to inform decisions on design. The manuscript would greatly benefit from the clarification and inclusion of key methodological details and justification for these as noted below:

Response to general comment

We wish to express our appreciation to the referee#3 for insightful comments on our paper. The comments have helped us significantly improve the paper. We have remade supplemental movie with indicator, subtitles, and commentary. We also have changed some section, particularly methods section, based on your comments.

Response to minor comments

Comment1

Line 173. Were all fish provided from the same ornamental traders, and of the same age? There could be baseline genetic differences between individuals which would need to be balanced across treatments or used as a blocking effect in the final statistical model.

Response

Thank you for your suggestion. We could not estimate age of fish individuals but they were almost same size. The fish were gain from some ornamental fish companies to ensure homogeneity in kin relationship between the subject male and presented fish. But we could not distinguish that they were sibling or non-relatives. Therefore, we did not include these factors in final model.

Comment2

Was any effort made to balance the size or age of fish between treatments/pairs/original stock tanks? Stocking density differed between the three holding tanks which could potentially have an impact on subsequent fish behaviour. Were efforts made to balance across the tanks so that there was equal opportunity to form a pair bond?

Response

We adjusted sex ratio (male: female = 1:1) each tank to control their reproductive motivation. We also controlled fish density. Moreover, no systematic bias in testing conditions. We have added the sentences to explain it. [L201-202]. Thank you.

Comment3

What happened to the pairs after pair bonding had occurred? Were they kept in separate tanks with ideal spawning conditions (e.g. clay pot as spawning substrate, ideal temperature?). Further, how was a successful pair bond defined? Itzkowitz and Bockelman (2008) (<https://doi.org/10.1163/156853908783402939>) provide examples of specific criteria used based on display of two courtship activities. This could influence the strength of the decision for prosociality in the subsequent test.

Response

Actually, we decided formation of reproductive pair by spawning event and their pair-specific display. Therefore, we have included a new section for definition of reproductive pair formation with new reference.

After spawning, fish pair were kept in same stock tank. But, because small juveniles will be predated by other fish, we removed them from stock tank when they grew.

Comment4

It may help to clarify that this species shows clear sexual dimorphism so that the male subject are capable of distinguishing between individuals or at least different sexes.

Response

Thank you for your suggestion. We have added the sentence for sexual dimorphism in this species. [L204]

Comment5

Line 187. Was the same experimental tank used for all subjects? Was the water changed between subjects to prevent a potential influence of circulating hormones released by previous subjects?

Response

In total, we had 10 experiment tanks. The water changed depending on pH and water qualities even in experiment period. Moreover, we cleaned the tank when it was between treatments. we Therefore, we think there was no effect of hormones.

Comment6, 7

Line 197. Using the time taken to finish all the food or habituate could be an interesting additional variable to indicate the personality or anxiety levels of each fish.

Another potential behavioural measurement could be the time the male subject took to make the decision, which could indicate a strength of decision making in addition to the binomial prosocial/antisocial.

Response

Thank you for your suggestion and comment. In this study, we focused here on testing whether this fish has prosociality. But, we are interested to the relationship between personality and intensity of prosociality. We will try to establish another experimental paradigm to reveal this question and think this is important matter for future studies.

Comment8

Line 204. Why was this determined as the end point for phase 2? I assume it is because they have learned to go to the doors to get the food.

Response

We thought that fish after learning relationship between door opening and feeding may will be tend to wait in front of door before door opening.

Comment9

Line 212. Were there differences between colour and shape (as shown in Fig 1.)? Which colours were used and have previous experiments shown that fish can differentiate between them or show any preference for a certain combination (e.g. certain colours could be innately associated with a preferred food source).

Response

Thank you for your suggestion. We did not refer the shape of mark in previous manuscript. Therefore, we have added the sentence. [L242-247]

Color and shape were completely randomized every experiment. And, we did not include these factors in analysis because mark meaning counterbalanced across subjects.

Comment10

Line 225. Were all trials conducted at the same time of day? I am unsure if these fish show any circadian rhythms!

Response

We conducted at the same time per 3 trials. It was ca. 9:00, 13:00, 16:00. We have supplemented the methods section with explanations of time. Thank you. [L257-258]

Comment10

Please clarify what is meant by a 'trial' and were there any gaps between trials? Was there any evidence of satiation by the end of the 9th trial for each day? Why was this number of trials chosen? Was it based on previous literature/a statistical power test/time limitations?

Response

We had to do more trials to gain the statistical power, but in our previous experiments, trials more than this made subjects to be obese and we thought it affects fish health. Therefore, we decided this experimental design. Trial means one choice, and was performed 9 times per day. We have changed the sentences in choice experiment section. [L256-257].

Comment11

Line 238. Are any of these behavioural measurements based on previous literature i. e. other fish choice tasks/showing mate preference?

Response

Thank you for your question. This experimental paradigm was inspired some primate works but not based on fish works.

Comment12

Line 229. A potential way of investigating side bias could have been to swap the location of colours randomly between trials for the same subject. This would provide a definite answer to whether the fish was showing side bias rather than making a consistent choice. Further, were the biased fish always showing preference for the same side? Could this be explained by an environmental effect?

Response

We agree with you. But we failed to experiment with swapping the location of coloration (or mark). We think it was difficult to establish the associative learning by subject. Subjects showing strong side bias always chose the choice nearby them. As your earlier comments, we think it is associated with subjects' personality or boldness.

Comment13

Was the collection of behavioural observations conducted by the same person? If so, were any efforts made to check intra-observer reliability?

Response

Behavioral observations were conducted by the same person under the blind condition. She is expert for behavioral observation but did not know our predictions. Therefore,

we did not test the intra-observer reliability.

Comments14

Line 309. For clarity it would help to state why this period was chosen. Is this based on the previous model' s results or on the previously stated hypothesis?

Response

We chose this period based on previous model' s results. Therefore, we have added the sentence to explain it. Thank you for your suggestion. [L337]

Comment15

S5 - I don' t think it is necessary or appropriate to include a direct screenshot of a statistical output.

Response

We have deleted one supplemental information. Thank you.

Comment16

Video - this is a good idea however it could be improved. The current video is short and of poor quality due to reflection and has no indicator of what it is showing. A longer video with subtitles/commentary would be more influential.

Response

We have remade the supplemental video with indicators, subtitles, and commentary. Moreover, we have included a new typical example video of quick turn behavior. Thank you for your suggestion.

Comment16

Error bars are present in the appropriate figures and defined - what statistical method used to generate these intervals is not defined. In Figure 2. the 95% CIs appear to be the same across all trials for the antisocial choice but differ between trials in the prosocial choice - is this correct?

Response

In figure 2, the 95% CIs in control experiment is similar but actually it is a little different among each day.

Comment17, 18

Figures 2 and 3 show no information on statistical significance or p values.

Figure legends are sparse - particularly Figures 2, 3 and 4, as well as the supplemental

data. More information is needed e.g. What means were used and from which model?

Response

Thank you for your suggestion; however, our statistical analysis included testing the 2-way interaction between factors. Therefore, we could not show p values in figure and their legends. But, we have added more information in legends of figures.

Comments19

Size appears to be an important factor in mate choice for these species, so I am surprised there is no mention of its consideration in this study, with both sexes preferring large mates. I would expect size to be used to balance between treatments and could also be used as covariate for the strength of the other-regarding preference. Size is likely to be an influential factor if these experiments were repeated the females as the subject, considering they will swap mates for a larger mate (see Gagliardi-Seeley, 2009 - 10.1007/s10164-008-0111-2).

Response

Pair formation between subject and their mate occurred in stock tank that mimicked natural condition. We think they chose mate from potential mates in stock tank therefore, there are no effect of mate choice in this study. In addition, we our subjects (and their mate) were similar body size. We have included a new information of body size in methods section. [L283-284]

Comments20

Reproductive status of the new female or current mate could be another factor to consider and have an influence on the male' s choice, particularly when both females were present. This could easily be quantified through used of the female' s 3 distinct colour phases, which correspond to their reproductive status

Response

Thank you for your comment. We used matured female which they experienced parental care with their mate. Therefore, we think that reproductive status of new females was controlled.

Comment21

The abstract is perhaps too concise in terms of information. Accessibility and clarity could be improved through defining what is meant by 'proactive prosociality' in the first sentence, what is briefly involved in the task and what is meant by a 'behaving prosocially' , i.e. eating out of side that meant the other fish also got

food.

Response

Thank you for your suggestion. Because of word limitation, we could not include more information in abstract. Instead of more information, we have changed “proactive” to “spontaneous” to explain more specifically.

Comment22

Line 55. Please clarify why only males were used as the test subject.

Response

Because female is smaller than male, we could not test using female as a subject using same experimental setup (females always chose the compartment nearby her). Therefore, we made different experimental setup for female choice experiment. But, we could not compare the results of two experiments because we used different experiment setup. Therefore, we decided that we try to submit by male only data.

Comment23

Line 62. It would help to repeat here what the control condition was.

Response

Thank you for your suggestion. We have redrafted this section [L66-67] to establish a clearer focus.

Comment24

Line 64. Provide a brief explanation of how this hypothesis was tested.

Response

We have included a new sentence how this hypothesis was tested. Thank you. [L69-70]

Comment25

Line 64-69. Are there any references from studies in other species to support the design of your test and logic of your hypotheses? The ‘foraging rate’ justification for adding these extra treatments is provided in the discussion but not here?

Response

We inspired to primate work in this study (de Waal et al. 2008). But consequence of choice of fish and primate is different. Therefore, we did not cite this reference here.

Comments 26

8. References: Does this manuscript reference previous literature appropriately? If not, what references should be included or excluded?

I feel it would be advisable for the authors to review other papers using this particularly animal model, such as those referenced in the comments above. This would help to fill in some of the information I feel is missing (again as noted above) to establish whether these factors have been overlooked during experimental design (e.g. reproductive status).

The overall conclusion is appropriate and clearly placed within wider hypotheses and discussed in the context of previous literature of the interdependence between the nature of reproductive relationships (e.g. the presence of parental care allocation) and the presence of prosociality.

Response

As mentioned above, we have added some new references. Thank you for your suggestion.

Comments 27

10. Please indicate any particular part of the manuscript, data, or analyses that you feel is outside the scope of your expertise, or that you were unable to assess fully.

The final statistical model reported would benefit from revision by a statistical expert. The authors have made a clear effort to take account of potential order effects and the unbalanced allocation of treatments. I feel I am not qualified to ascertain the appropriateness of this model and whether there are sufficient or valid replicates.

Response

Throughout this review, we have tried improvement for the visibility of statistical analysis. Now, our manuscript will be more sophisticated. Thank you very much.

Again, thank you for giving us the opportunity to strengthen our manuscript with your valuable comments and queries.

Referee#4

This simple, but elegant experimental study shows how proactive (but cost-free) prosociality is linked to the presence of strong social bonds and parental care relationships, and thus provides strong support for the interdependence hypothesis for proactive prosociality. The study controls for the possibility that the prosocial acts are reactive, and that the subjects simply liked seeing more food around (by adding a rival as possible recipient). Overall, this is a very convincing and well-designed study. I can find nothing major to object or question, and it would make a valuable addition to the literature. I must admit that I am not a fish expert, so cannot judge whether the natural history information on this species is presented correctly (but it seems to me it is).

A second attractive feature of the paper is that the results suggest an awareness of the risk of partner loss when alternative partners are presented in view of the current long-term partner. This result fits neatly into the wave of recent studies showing so far unexpected cognitive abilities in fishes.

I cannot find much to object to, but hope the authors can reassure me that the exclusion of individuals with a persistent side bias (P9, 1229) cannot have affected the results that were obtained.

Response to general comment

We appreciate your compliments. We also wish to express our strong appreciation to the reviewer#1 for their insightful comments on our paper. We feel the comments have helped us significantly improve the paper. We hope that we can solve your worrisome point of side bias through this response.

Response to minor comments

Comment1

P2, 134 It is fair to say these cognitive abilities are necessary for the findings reported in this paper, so merely saying that they have been shown in some fishes is, strictly speaking, not enough: they must be demonstrated to be present in the species that was tested. This requirement should not sink the paper, but some mention of this assumption is required.

Response

Thank you for your suggestion. We have added the new reference of individual

recognition (familiar-unfamiliar recognition) in introduction section. [L45-47]

Comment2

P3, 138-43 These lines do not belong in the introduction, but in the discussion, since they have no bearing on the predictions.

Response

We have deleted this section from introduction and then insert conclusion section. Thank you very much. Our prediction and discussion will be more sophisticated by your suggestion. [L189-192]

Comment3

P3, 155 It would be helpful to explain why only males were tested.

Response

We tested using female fish as a subject but female always chose compartment nearby her and therefore showed strong side bias. We think it is caused female is smaller than male and can not recognize two compartments. Therefore, we showed data of males only in this study.

Comment4

P4, 182 The monkeys behaviorally examined the other-regarding preference. What does that mean? If they mean whether donors checked whether recipients took the food or were interested, in other words whether they intentionally produced food for their mates, this has been studied rigorously more recently (see Burkart & van Schaik 2020, Anim Cogn).

Response

Thank you for your suggestion. We have changed this section and included reference you suggested. [L90 and L110-111]

Comment4

P5, 1109-10 I do not understand "which is indeed often the compartment" Please clarify.

Response

Sorry it is our mistake. We have changed this sentence. [L109-119]

Comment5

P5, 1120 I did not see an attempt to explain the minor differences in the three

treatment order groups as a function of differential experience or the order in which the experience was gained. Perhaps this could be added to the supplementary information.

Response

You have raised an important point; however, we do not have idea of discussion for order effects for prosociality because of lacking information and data. We think we must invent new experimental systems to test the order effects for their prosociality.

Comment6

P6, 1137+ One would expect an effect of the time elapsed since the male was separated from his mate on the tendency to provision a strange female. This could easily be checked in the data.

Response

Thank you for your suggestion. In our experiments, time elapsed since the male was separated from mate was same every experiment. Therefore, based on your suggestion, we have included a new supplemental figure [supplemental Figure S6]. In this figure, we showed difference on the prosocial choice between “mate” and “new female alone” experiments during ten days. We hope this alternative solve your indicated item.

Comment7

P9, 1229 I do not understand how fishes can have a side bias when during the whole experiment the food compartment is always one and the same. This is the only aspect of the study that I find worrisome, so please explain.

Response

As figure 3b, we had two compartments. Fishes showing side bias always stayed left or right side of compartment and chose compartment nearby him regardless of consequence of choice. We have attached image to explain this. We hope that we can solve your worrisome point.

Comment8

P10,259 potential rather than potentially, unless an adjective is missing after potentially.

Response

We have changed it. Thank you for your suggestion. [L290]

Comment9

P12, 1312+ Why are these analyses done on the aggregate data rather than on a per-trial basis?

Response

We are sorry it's hard to understand. As you said, we analyzed this data per trials, namely; we included trial identify as a random factor in the model. We have changed this sentence. [L340]

Comment10

Finally, in various places, the authors refer to 'active' prosociality, which blurs the distinction between proactive and reactive. So, please either define it or use the terms proactive and reactive.

Response

Thank you for your comments. We have changed "active" to "proactive" and unified consistent wording throughout.

Again, thank you for giving us the opportunity to strengthen our manuscript with your valuable comments and queries.

Reviewer #1 (Remarks to the Author):

I would like to thank the author for their thoughtful comments and thorough revisions. I feel that all of my concerns have been addressed. Given the importance of this study to the field, I sincerely hope that it is accepted for publication.

Reviewer #2 (Remarks to the Author):

This new version of the manuscript is much improved, and I am happy to report that the authors have successfully addressed the points I raised during the previous round of review, and therefore to recommend this paper for publication.

I have 3 more additional minor points:

Lines 135-136: Given the fact that the authors have no information about the lineage of these fish and they therefore could not directly test for this, I would suggest removing this sentence

Lines 175-177: As there is no direct exploration of the neuroanatomical basis of the observed behaviour, I think that the parallel to the similarity between the social decision-making network of fish and endotherm vertebrates should be removed. This study does indeed show that the cognitive processes are probably similar, but provides no new evidence to support that the physiological mechanisms underlying this process are similar.

Line 287: I suggest rephrasing to "376 data points"

Reviewer #3 (Remarks to the Author):

I am happy that the authors have made sufficient changes in response to detailed reviewer's comments.

Reviewer #4 (Remarks to the Author):

I found the authors' responses convincing.

Response to reviewers

Response to reviewer#1

Comment

I would like to thank the author for their thoughtful comments and thorough revisions. I feel that all of my concerns have been addressed. Given the importance of this study to the field, I sincerely hope that it is accepted for publication.

Response to comment

Thank you for all your cooperation for our manuscript. Our manuscript will be improved by your critical comments.

Response to reviewer#2

General comment

This new version of the manuscript is much improved, and I am happy to report that the authors have successfully addressed the points I raised during the previous round of review, and therefore to recommend this paper for publication.

Response to comment

We really appreciate your help in our reviewing for this manuscript.

Minor comment1

I have 3 more additional minor points:

Lines 135-136: Given the fact that the authors have no information about the lineage of these fish and they therefore could not directly test for this, I would suggest removing this sentence

Response to comment

We are sorry that our sentence was difficult to understand. We thought kin selection can be excluded because we used unrelative fish as subject and presented fish. To reveal our opinion, we have changed this sentence. [L169-171]

Minor comment2

Lines 175-177: As there is no direct exploration of the neuroanatomical basis of the observed behaviour, I think that the parallel to the similarity between the social decision-making network of fish and endotherm vertebrates should be removed. This study does indeed show that the cognitive processes are probably similar, but provides no new evidence to support that the physiological mechanisms underlying this process

are similar.

Response to comment

As you said, we did not have data for their physiological mechanisms. Therefore, we removed this sentence. [L212]

Minor comment3

Line 287: I suggest rephrasing to “376 data points”

Response to comment

We have changed “376 data” to “346 data point” . [L322]

Thank you again for everything you’ ve done.

Response to reviewer #3

I am happy that the authors have made sufficient changes in response to detailed reviewer’s comments.

Response to comment

We cannot thank you enough for helping us.
Our manuscript was sophisticated by your comments.

Response to reviewer #4

I found the authors’ responses convincing.

Response to comment

Thank you for your kind cooperation.
Our manuscript was sophisticated by your comments.